# Design, Fabrication and Evaluation of a Stretchable High-Density Electromyography Array

**DOI:** 10.3390/s24061810

**Published:** 2024-03-11

**Authors:** Rejin John Varghese, Matteo Pizzi, Aritra Kundu, Agnese Grison, Etienne Burdet, Dario Farina

**Affiliations:** Department of Bioengineering, Faculty of Engineering, Imperial College London, London W12 0BZ, UK; r.varghese15@imperial.ac.uk (R.J.V.); matteo.pizzi19@imperial.ac.uk (M.P.); a.kundu@imperial.ac.uk (A.K.); agnese.grison16@imperial.ac.uk (A.G.); e.burdet@imperial.ac.uk (E.B.)

**Keywords:** wearable sensors, human–machine interfacing, surface electromyography, HD-EMG, soft robotics and sensing

## Abstract

The adoption of high-density electrode systems for human–machine interfaces in real-life applications has been impeded by practical and technical challenges, including noise interference, motion artefacts and the lack of compact electrode interfaces. To overcome some of these challenges, we introduce a wearable and stretchable electromyography (EMG) array, and present its design, fabrication methodology, characterisation, and comprehensive evaluation. Our proposed solution comprises dry-electrodes on flexible printed circuit board (PCB) substrates, eliminating the need for time-consuming skin preparation. The proposed fabrication method allows the manufacturing of stretchable sleeves, with consistent and standardised coverage across subjects. We thoroughly tested our developed prototype, evaluating its potential for application in both research and real-world environments. The results of our study showed that the developed stretchable array matches or outperforms traditional EMG grids and holds promise in furthering the real-world translation of high-density EMG for human–machine interfaces.

## 1. Introduction

Electromyography (EMG) records the electrical activity produced by skeletal muscles during a contraction. Such activity is triggered by the firings of alpha motor neurons (MNs) in the spinal cord, which are regulated by supra-spinal commands from the motor cortex, and influenced by spinal mechanisms of excitation, inhibition, and feedback [1]. In interfacing applications, EMG is a source of information about the user’s motion intent [2,3,4]. Because of its ease of use, high-signal quality, and stability, EMG has so far delivered more encouraging results compared to other approaches [5].

EMG human–machine interfaces (HMI) have traditionally been used for neuro-rehabilitation [6], control of prosthetics or exoskeletons [7,8], motor enhancement [9], or basic motor control investigations. The further transition of these devices to healthy consumer applications appears imminent, although it is challenged by practical and technical difficulties. Shortcomings range from noise and motion artefacts [10] to the complexity of the algorithms needed to decode multiple degrees-of-freedom (DOFs) simultaneously [11], and the lack of small and compact electrode interfaces [12]. In laboratory settings, practical challenges include ensuring wearability, the need for portable biosignal amplifiers with extensive channel capacity while managing cable bulk, and durable electrode interfaces with cross-user adaptability [13,14,15]. Current solutions used in research predominantly involve wet-electrode grids, which require adhesive foams to ensure continuous contact between the electrodes and the skin, as well as conductive pastes to reduce skin–electrode impedance [13]. In addition, surface preparation is required to maximise the signal quality. This usually involves shaving the skin area where the electrodes are placed, rubbing the skin with an abrasive paste to remove dead skin cells, and ultimately cleaning the surface with alcohol [14]. Additionally, most of these electrodes are fixed in place using tapes and/or bandages. This procedure is time-consuming, expensive, sometimes unpleasant, and an impediment to scalable real-world translation [15].

To cope with these issues, dry-electrodes have been used for different applications [16]; however, available solutions still lack an adaptive and easy-to-use application process [17]. While HD-EMG stretchable sleeves (Battelle Memorial Institute, Columbus, OH) [18] with dry electrodes exist, their accessibility is limited. Furthermore, replicating the fabrication process of this array sleeve is time-consuming and requires access to specialised equipment and expertise to integrate electronics within fabric substrates. Additionally, the latter may suffer from challenges related to cleaning or adherence to hygiene standards.

In this paper, we introduce a wearable high-density EMG (HD-EMG) stretchable array (Figure 1), detail its design and fabrication methodology, and compare its performance to commercially available grids, validating it for gesture recognition tasks and EMG signal decomposition. The developed device can be fabricated relatively easily, with potential for the fabrication strategy to be applicable both in research and scale for real-world applications. The design of the stretchable array is based on electrodes on a flexible printed circuit board (PCB) substrate. The fabrication process then allows us to convert the substrate into a stretchable interface. While we present this method through the development of an HD-EMG sleeve, the technique can generalise to other modalities, ranging from wearable ultrasound and inertial measurement units (IMUs) to vibration motors and functional electrical stimulation (FES) electrodes. With the presented fabrication method, we present the community with a DIY method that would allow researchers and makers to fabricate stretchable human–machine interfaces that would be wearable, portable, and generalisable, while ensuring hygiene and minimising subject preparation constraints.

## 2. Design and Fabrication

### 2.1. Fabrication

The stretchable array presented in this section was conceptualised to overcome the practical challenges experienced while using commercially available HD-EMG grids, and facilitate realisation of a more practical setup while ensuring comparable signal quality [20]. The fabrication method was developed to be executed within a lab/maker-space setting using materials that can be easily purchased (such as the flexible PCB grid and silicone rubber) or fabricated using 3D-printing/laser-cutting (such as the moulds). The materials and fabrication method, which evolved as different prototypes were realised (Figure 2B), are described in the following and graphically presented in Figure 2A.

1.A thin soft layer (layer A) is created by pouring the silicone rubber in mould A.2.Soft layer A is inlaid in mould B, and the flexible PCB is then placed on top of layer A, by folding the sections between electrodes, and by press-fitting these into the holes of mould B. This step ensures that the flexible PCB maintains its compressed configuration during fabrication.3.The rear of the flexible PCB is then covered by a layer of fabric which is held in place by mould C. Mould B and C are aligned with indexing pins and tightly held in place using clips.4.The silicone rubber is poured evenly on top of the fabric and is allowed to seep in through the same and the gaps within the folds of the flexible PCB and the moulds.5.Post-curing, the entire stretchable array can be removed as a single unit. Fastening, such as Velcro straps, press-buttons, etc., can be provisioned either on the cured rubber or the excess fabric to allow it to be fastened around the arm.

The above fabrication process results in the stretchable array shown in Figure 2 and Figure 3, and can be used either as a closed sleeve or as an open array/patch. The prototype also allows for multiple arrays to be combined together with the fasteners in series or parallel to create bigger arrays. All files (including PCB schematics, mould computer aided design files (CADs), etc.) have been uploaded on GitHub (https://github.com/rejinjohnvarghese/Stretchable-HMI-Array, accessed on 4 February 2024).

### 2.2. Materials and Component Choices

#### 2.2.1. Electrodes

Electrodes (Dongguan Zhongxin Precision Technology Co. Ltd., Dongguan, China) with a copper base and an Electroless-Nickel-Immersion-Gold (ENIG) coating (3μm layer of Nickel and 0.04μm layer of Gold) were used. The coating ensures low impedance and high electrical conductivity to maximise signal quality [21]. Each electrode was ∅10mm and 1.5mm tall. While any electrode size is compatible with the fabrication method, the choice of electrode diameter was made to use the array dry without the need for excessive surface preparation or conductive paste [22].

#### 2.2.2. The Flexible PCB Grid

A flexible PCB was used as the base substrate for the stretchable array. As the stretchable array was conceived, primarily, to be worn around the forearm, the length of the folded and original configuration of the flexible PCB was determined from the average forearm circumference. Given the lack of stretchability in a flexible PCB, a range of 240–320 mm achieved between the folded and original (resting) configuration, respectively, allowed us to broadly cover a statistically significant population of both male and female adults. To achieve the above range of measurements, and given the choice of electrodes (∅10mm), a 16×4 configuration was chosen to maximise the effective measuring area, resulting in an inter-electrode distance of 14mm (sleeve in unstretched configuration, flexible PCB in folded configuration). A staggered arrangement of electrodes along the width of the array allowed for the same to be minimised to ∼42mm, as stretchability was not considered necessary along this axis. KX14-70K8DE connectors (JAE Electronics, Tokyo, Japan) were used to ensure compatibility with commercially available amplification systems. The base material of the flexible PCB was flexible polyimide with a thickness of 0.1mm (authors recommend using thicker substrate for improved robustness), and was fabricated by PCBWay (www.pcbway.com, Hangzhou, China).

#### 2.2.3. Silicone Rubber Substrate and Fabric Reinforcement

To allow for stretching and compression of a material that is inherently not flexible, the flexible PCB was folded and encased within a stretchable substrate (described in the next subsection). Also, using a silicone-based material allows for improved cleaning and hygiene over fabric-based alternatives. The silicone rubber used for the sleeve needs to have low viscosity (high pourability) to ensure the material flows between the moulds and folds of the flexible PCB, and results in low air entrapment while mixing, reasonable curing times, strength and low degradability post-curing. Magic Rubber (Raytech S.r.l, Milan, Italy), a fast cross-linking insulating bi-component liquid rubber was used for the prototypes.

The strains encountered between the default (240mm) and fully stretched (320mm) configurations is within the acceptable range of strains that silicone rubbers can withstand. However, to prevent fatigue failure of the silicone rubber due to repeated and considerable stretching of the array, reinforcement with an equally stretchable yet porous and strong material (such as PowerMesh fabric) was employed to improve the strength, structural integrity, and durability of the prototyped array. Conductive stretch fabric (Kitronik Ltd., Nottingham, UK) was also investigated to limit the effects of external electromagnetic interference (EMI), by recreating a Faraday cage. This did not result in significantly improved results so far, but will be investigated further in the future.

## 3. Characterisation Experimental Methods

### 3.1. Baseline Noise Characterisation

To compare the noise performance of the electrode grids, a baseline-noise characterisation was conducted by computing the average root mean square (RMS) of each of the 64 channels, for the developed dry-electrode grid. As a reference metric, the same test was also run on an analogous grid, for which each electrode was covered by a layer of conductive gel (this type of grid will be referenced as ‘wet-electrode grid’ from now on). Both recordings were taken using the MyoLink amplifier [19], after filtering signals with a 50 Hz notch filter, and 10–500 Hz bandpass filter [23]. In particular, the subjects were asked to maintain a ‘rest position’ (no movement nor force produced), while the baseline signals were recorded for 10 s, with the grids placed along the TA muscle (Figure 3iii). After placing the first grid on the TA muscle, the accurate position (on the subject’s leg) was marked using a skin marker. This was done to allow proper placement of the wet-electrode grid afterwards, and thus a fair comparison between the resulting RMS values. The average RMS was then computed across all channels using Equation (Equation 1):(1)$${\mathrm{RMS}}_{overall}=\frac{1}{N}\sum_{n=0}^{N-1}\sqrt{\frac{1}{T}\sum_{t=0}^{T-1}x{\left(t\right)}^{2}}$$
where *N* is the total number of channels (64 for one grid), and *T* is the total number of data samples recorded.

A statistical analysis was first conducted to discard the outlier channels (e.g., particularly affected by noise, likely due to physical damage of the electrodes or poor contact with the skin). This was performed by computing the Z-score value ($Z_{score}=\frac{RMS_{value}-\mu }{\sigma }$) for each channel, and removing the ones for which it was >3 or <−3 (values further than three standard deviations from the mean).

Furthermore, to assess the significance of the obtained results, a two-sample *t*-test was used to compare the RMS values between the two types of electrode grids. For this, the null hypothesis was set to be ‘no significant difference between the RMS values of the two groups’, while the alternative hypothesis would be ‘significant difference between the RMS values of the two groups’. The *p*-value to determine the significance was set to be 0.05.

### 3.2. Electrochemical Characterization of Electrode Sites

To measure the impedance and charge transfer of the electrode sites of the sleeve electrode, Electrochemical Impedance Spectroscopy (EIS) [24] and Cyclic Voltammetry (CV) [25] were performed using a potentiostat/galvanostat (PGSTAT101, Metrohm AG, Herisau, Switzerland). The Gold-coated (n=6) electrodes were submerged in phosphate-buffered saline at room temperature. The adopted characterisation protocol follows the method presented in [26]. A three-electrode configuration with EMG electrodes as working electrode (WE), a platinum counter electrode (CE), and an Ag/AgCl as reference electrode (RE) was used. Impedance spectra were characterized between 0.1 Hz and 100 kHz by applying a sine wave of 30 mV. The CV was calculated over 11-step cycles. Each measurement (EIS and CV) was repeated three times. The mean for the EIS and CV were calculated.

## 4. Validation Experimental Methods

### 4.1. EMG Model

EMG signals provide valuable insight into the dynamics of neuromuscular systems by capturing the electrical potential fields generated during muscle contractions. These contractions are a result of the spiking activity of motor units (MUs) [27]. As direct recording from the MU soma or axons is invasive and impractical, analysing EMG signals has become the preferred method for understanding the neurophysiological activity of MUs [28,29]. The EMG signals we record are the result of the spatial and temporal summation of potential contributions from all activated muscle fibres, irrespective of their originating MU [27]. Conceptually, this aggregate signal is the cumulative summation of motor unit action potential (MUAP) trains. A significant challenge in working with EMG signals is the decomposition of these composite recordings to isolate and analyse the activity of individual MUs and their firing patterns. Mathematically, EMG signals can be represented as a convolution of sources:(2)$$x\left(t\right)=\sum_{l=0}^{L-1}H\left(l\right)\tilde{s}\left(t-l\right)+n\left(t\right)$$
where $x\left(t\right)={\left[x_{1}\left(t\right),x_{2}\left(t\right),\ldots ,x_{m}\left(t\right)\right]}^{T}$ is the vector of *m* EMG signals,
$\tilde{s}\left(t\right)={\left[s_{1}\left(t\right),s_{2}\left(t\right),\ldots ,s_{n}\left(t\right)\right]}^{T}$ are the *n* MU spike trains that generate the EMG signals, and n(t) is the additive noise. The matrix H(l) with size m×n contains the *l*-th sample of the MUAPs for the *n*-th MU and the *m*-th EMG observation. The challenge then becomes separating the individual sources from the mixed signal. To retrieve the sources from the observations without any a priori information, also known as a Blind Source Separation (BSS) approach [30], several algorithms have been proposed. A notable example of automatic decomposition for neurophysiological signals can be found in the convolutive BSS method [31]. In this approach, FastICA is utilized to refine the separation vector, leading to an efficient convergence to the source. FastICA has been previously employed for EMG decomposition, and the decomposition results are consistent in measurements separated by weeks [32].

### 4.2. Data Acquisition

Validation experiments for the proof of concept of the sleeve were performed on the forearm muscles of two participants (both males, of 23 and 22 years old) and the Tibialis Anterior (TA) muscle of one participant (male, 23 years old). Both the dry and wet configurations of the sleeve were tested under the same conditions (Section 3). In both cases, the EMG signals were acquired in monopolar derivation [33]. All experimental procedures adhered to the ethical guidelines set by the Imperial College Research Ethics Committee (ICREC) under approved application 22IC7655.

#### 4.2.1. Recording Forearm EMG During Gestures

The EMG signals for the forearm experiments were recorded with the MyoLink, which is a portable, modular, and low-noise electrophysiology amplifier [19], sampled at 2000 Hz, and A/D converted with 24-bit resolution. A reference electrode was placed on the wrist. The subjects maintained a seated position while recording the gestures, with the arm parallel to the torso, and the forearm perpendicular to it. The elbow was positioned on a movable support, to prevent excessive fatigue during the experiment sessions. This position was chosen, as it could be easily standardised across subjects, and would allow for an accurate repetition of the gestures. In addition, the sole purpose of this validation experiment was to compare the accuracy of the dry-electrode configuration, against the wet one, and not assessing how the recognition accuracy would change according to the subject’s position. Participants were instructed to replicate seven dynamic wrist gestures—flexion, extension, radial deviation, ulnar deviation, pronation, supination, and hand closing—following a visual cue. The cue was structured to allow 3 s for adopting the gesture, 10 s for holding it, and 3 s to return to the rest position. Six trials were recorded for each gesture. For each trial, the four electrodes closest to the Velcro strap (below the connector) were removed since they were obstructed by the overlapping ends of the sleeve (when wrapped around the arm), and skin contact was not optimal.

#### 4.2.2. Recording TA EMG during Isometric Contraction

The EMG signals were recorded with a Quattrocento amplifier (OT Bioelettronica, Torino, Italy), with 16-bit resolution, sampled at 2048 Hz and bandpass filtered between 10 and 900 Hz. A reference electrode was placed on the ankle (TA). The participants were instructed to sustain isometric ankle dorsiflexion (Dynamometer, OT Bioelettronica, Torino), with the force displayed as feedback and concurrently recorded with the EMG signals. The relative force was determined as percentages of the maximal voluntary contraction (MVC) and was set to be sustained at 20% MVC. Moreover, the same seven electrodes detected as ‘outliers’ during the Z-score analysis run for the baseline-RMS experiment (Section 3.1) were discarded as well.

### 4.3. Model Architecture and Parameters

*Classification Model*: A simple convolutional neural network (CNN) taking the raw-EMG signal as input (with architecture illustrated in Figure 4) was used to perform the classification across the seven classes indicated in Section 4.2.1.

Each layer is followed by the ReLU activation function (ReLU(x)=max(0,x)), except for the last dense layer, which uses the Softmax function, defined as $\sigma {\left(z\right)}_{j}=\frac{e^{z_{j}}}{\sum_{k=1}^{K}e^{z_{k}}}$. The kernel size and stride for the convolutional layer were set to (50, 5) and (10, 3), respectively, while the pool size and stride of the max pooling were both set to (3, 1).

In particular, the isometric part of the gesture (10 s maintaining the hand and wrist in the same position) was extracted from each recording, and the resulting segment was split into smaller intervals of 250 ms. 80% of the obtained segments were used for training, and the remaining 20% for testing (the separation was performed using the train_test_split function provided by the sklearn library, shuffling the segments before performing the split). Before training and testing, each EMG signal was digitally filtered with notch filters to remove the 50 Hz, 100 Hz, and 150 Hz components, as well as using a digital bandpass filter between 20 Hz and 500 Hz.

*Performance Metric*: The network was trained for 20 epochs, using the Adam optimizer and sparse categorical cross-entropy loss for the objective function. Five-fold cross-validation was employed for each experimental condition, yielding the accuracy metrics for each fold.

### 4.4. Decomposition

The recorded data were decomposed with the open-source software MUEdit [31] for 200 iterations, with the silhouette cut-off set to 0.85 as the metric for assessing the quality of the source. Twenty-second intervals of the isometric hold phase were decomposed for analysis, separately for the dry and wet electrode configurations. After the decomposition, the data were automatically cleaned by selecting the MUs with a Coefficient of Variation CoV< 30 and a firing rate FR< 35 Hz [32].

## 5. Results

### 5.1. Baseline Noise Characterisation

After running the statistical analysis, a total of 57 electrodes (out of 64) was used for the baseline-noise characterization, which resulted in average RMS values of 14.55 μV and 13.5 μV (across all channels) for the dry and wet-electrode configurations respectively. This indicates an average 7.2% decrease in baseline RMS across all channels for the wet-electrode configuration, which is expected as the conductive gel reduces impedance and improves noise performance. Such result is supported by the two-sample *t*-test, which returned a T-statistic of 2.39, and a *p*-value of 0.018 (<0.05), indicating the rejection of the null hypothesis (no significant difference between the RMS values of the two groups), and the validity of the alternative hypothesis (significant difference between the RMS values of the two groups).

### 5.2. Electrochemical Characterisation

Figure 5A shows the impedance plots for the gold electrodes. The plots follow the standard metallic-coated character. Gold electrodes due to their large surface and high conductivity exhibited a well-defined capacitive behaviour acting like a low-pass filter. As seen in Figure 5A(i), gold maintained superior charge transduction above the characteristic frequency of 1 kHz. This can be attributed to the higher electrical conductivity of gold, which results in a less resistive system once the capacitive behaviour is overcome. The phase spectra (Figure 5A(ii)) of the materials revealed complex charge transfer behaviours. Gold electrodes exhibited a roughly −60° phase shift around 30 Hz, which corresponds to their capacitive behaviour. This phase shift is due to the accumulation of charge at the electrode/electrolyte interface as the frequency decreases, requiring more potential to overcome the attraction between negative and positive charges, leading to a delay in the current signal. As the frequency drops and the phase shift approaches the theoretical −90° threshold, the potential and current waveforms start to synchronize due to their sinusoidal nature. However, a decline in the phase angle was observed at both low and high frequencies. The low-frequency response can be attributed to the gradual influence of double-layer charging, like what happens with gold. At high frequencies, the relatively low charge carrier mobility of the conductive polymer may cause small delays in current, resulting in a decrease in the phase response above 10 kHz. Cyclic voltammetry (Figure 5A(iii)) measurements also highlight the typical metallic behaviour [25].

### 5.3. Experimental Validation

#### 5.3.1. Gesture Classification

Table 1 reports the average accuracy resulting from the 5-fold cross-validation, for each type of electrode grid and each subject. Figure 5B(i) shows a representative EMG signal recorded during a gesture recognition task.

The accuracy results over the seven gesture classification tasks show that both types of electrode grids perform well when validated on test data, with all accuracies reported above 95.9%. The wet-electrode grids slightly outperform the dry ones by a few percentage points, given the superior noise performance which results in signals with better signal-to-noise ratios. In particular, the wet-electrode grids improved the accuracy by an average of 1.59%. More specific classification accuracies for each gesture, subject, and electrode type are reported in the confusion matrices in Figure 5C(i–iv). Both the dry and wet electrode grids performed well on the test data, predicting the correct gestures in most trials.

#### 5.3.2. Decomposition

In the conducted decomposition experiment, a comparative analysis between dry and wet electrode grids was performed to assess their efficacy in detecting motor units. The results revealed that both electrode configurations exhibited a comparable performance, with 13 MUs detected using the wet-electrode grid, and 12 MU using the dry one. The average silhouette (SIL) value for the dry grid was measured at 0.922, slightly higher than the wet grid’s value of 0.918. Furthermore, the dry electrode grid demonstrated an average coefficient of variation (CoV) of interspike intervals of 22.8%, while the wet grid exhibited a lower value of 20.4%. In terms of average discharge rate, the dry grid recorded 12.46 Hz, slightly lower than the 12.67 Hz observed with the wet grid. These findings suggest that both configurations yield similar outcomes, with subtle variations in SIL, CoV, and discharge rates, likely due to the use of conductive gel in the wet grids, which improved impedance and noise performances. Figure 5B(ii) shows a representative EMG signal recorded from the TA during the isometric contraction at 20%MVC (dry-electrode configuration).

## 6. Discussion

### 6.1. Grid Design

This paper presents the design, fabrication method and experimental evaluation of a stretchable HD-EMG array, and more generally, presents the fabrication method for designing cost-effective stretchable human–machine interfaces. The proposed sleeve design allowed for instant fitting, improved comfort and wearability, easy adaptability to arms of different sizes, while reducing subject preparation needs. The chosen materials have high strength and resistance in comparison to commercial alternatives while ensuring hygiene, making it more suitable for real-world applications. The high-quality electrode materials resulted in low-noise and high-fidelity EMG signals, making the sleeve ideal for research settings as well.

### 6.2. Gesture Classification

As observed in Section 5, all electrode grids performed well in the gesture classification tasks, with the wet-electrode configuration marginally outperforming the dry configuration by 1.61%. Despite the expected marginal improvement, the trade-off from the perspective of real-world translation makes the wet-electrode solution impractical. Conversely, the dry-electrode configuration, with its quick application, and no need for skin preparation and gel application, is suitable for such conditions. In addition, the dry-electrode grid still achieved good results in the classification task, with the lowest test-set accuracy of 96.5%. These findings suggest the possibility of application of EMG-based interfaces for real-world scenarios.

### 6.3. Decomposition

The obtained results show that the dry-electrode configuration performs as well as the wet one, indicating that the type of electrodes used and the robustness of the assembly process help overcome the wet/dry difference between the two configurations, making the dry grid simpler, easier and more convenient to use. In terms of comparison with other studies focusing on MU decomposition from the TA muscle, the proposed design achieved comparable results in terms of number of extracted MUs [34,35]. This is attributed to the larger electrode size and inter-electrode distance of the proposed configuration. In particular, with respect to the grids more-commonly used for decomposition (wet HD-EMG grids); despite the lower spatial resolution, our design can achieve an overall lower impedance (due to greater electrode-skin contact area). In addition, the greater inter-electrode distance implies that after positioning the sleeve over the TA, the 64 electrodes would cover a larger area of the muscle.

## 7. Conclusions

This work proposes the design and fabrication method of a stretchable high-density, dry-electrode EMG array from a flexible PCB grid. The described manufacturing process can be completed with equipment available in most research laboratories (e.g., 3D printers, laser cutters) and the designs of both the flexible PCB and manufacturing moulds, along with all the parts and materials required for the assembly, have been made available open-source (https://github.com/rejinjohnvarghese/Stretchable-HMI-Array, accessed on 4 February 2024). The validation experiments demonstrated the quality of the electrode materials, resulting in low-noise and high-quality EMG signals. The design of the sleeve allows for stretchability and robustness, with high versatility in a variety of experimental conditions. In particular, the sleeve was benchmarked in experiments involving signal decomposition and hand-gesture recognition, always yielding an average accuracy of more than 95.9% across all seven gestures.

## Figures and Tables

**Figure 1 sensors-24-01810-f001:**
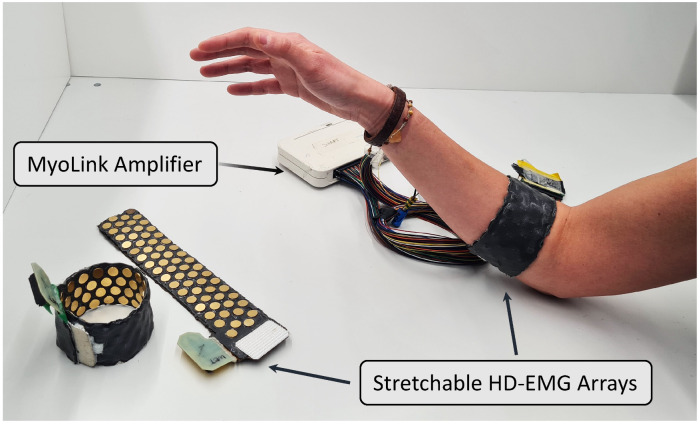
A setup using the developed stretchable HD-EMG arrays with the MyoLink amplifier [19].

**Figure 2 sensors-24-01810-f002:**
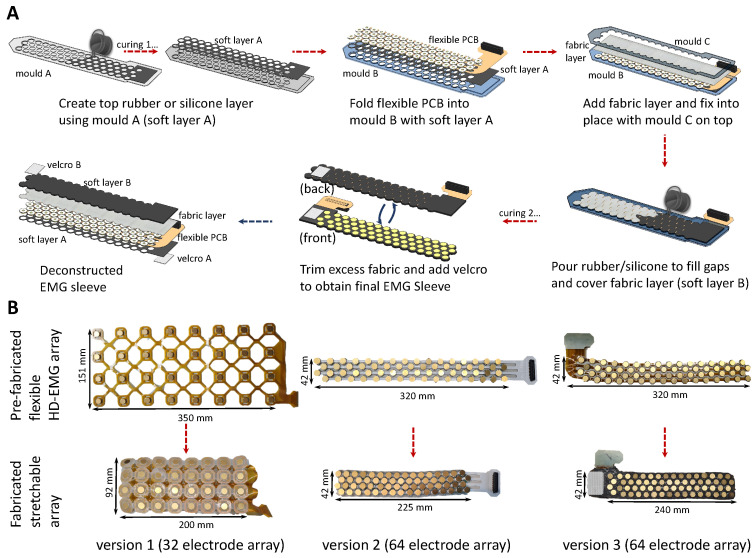
An overview of the fabrication methodology and evolution of the stretchable HD-EMG array. (**A**) The figure presents the fabrication and assembly of the different layers to realise the stretchable array from the flexible HD-EMG array. (**B**) The figure presents the evolution of the sleeve from a 32-channel array made up of only a single silicone layer to a 64-channel array. Version 3 benefits from a 4× electrode coverage due to improved electrode placement and benefits from superior stretchability and structural integrity due to construction reinforced with fabric and silicone.

**Figure 3 sensors-24-01810-f003:**
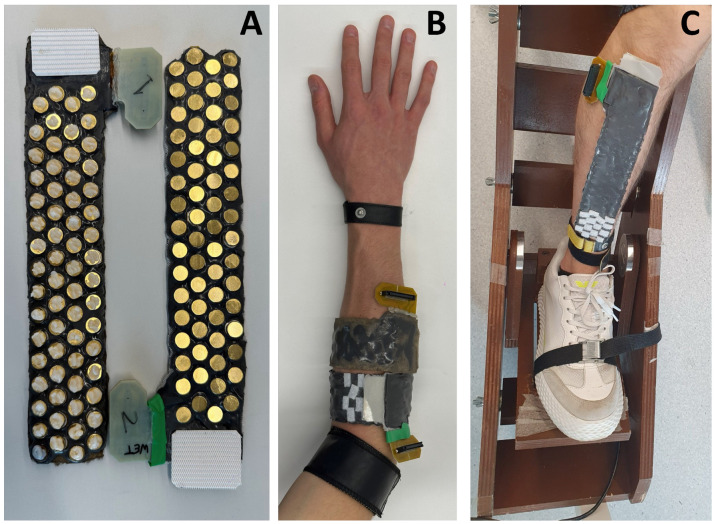
(**A**) Stretchable array in wet (left) and dry (right) configurations. (**B**) Experimental setup for the proof of concept for the dynamic recordings (Section 4.2.1). (**C**) Experimental setup for comparing the wet vs. dry grids on the decomposition of surface EMG signals (Section 4.2.2).

**Figure 4 sensors-24-01810-f004:**
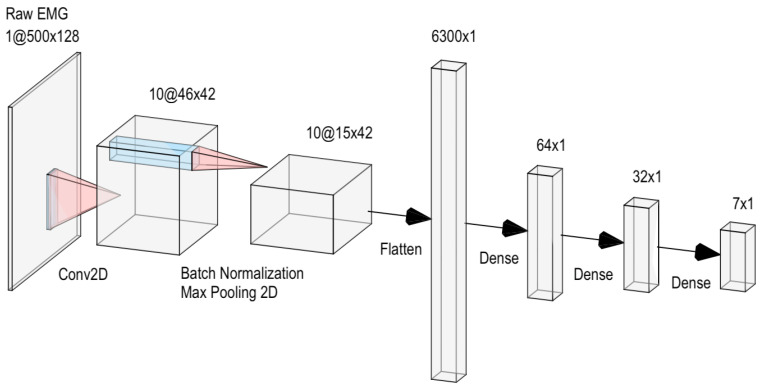
CNN architecture used for gesture classification.

**Figure 5 sensors-24-01810-f005:**
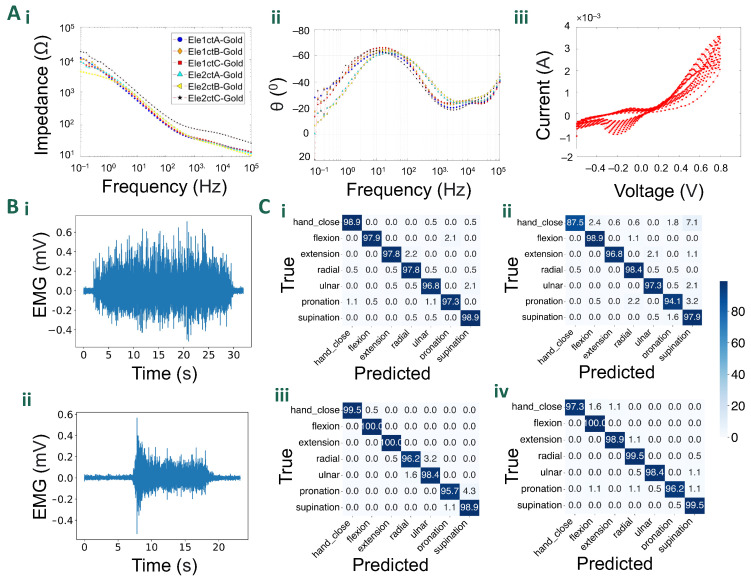
Characterisation and validation results: (**A**) Electrochemical characterisation results: (i). The impedance plot for gold-coated electrodes. (ii). The plot for the phase-angle for gold-plated electrodes. (iii). Cyclic voltammograms of gold-coated electrodes over eleven cycles. (**B**) Example EMG signal from a single electrode during (i). contraction of TA muscle for the decomposition task, and (ii) hand squeeze (both recorded using the dry-electrode configuration). (**C**) Confusion matrices from gesture classification validation experiments on subjects 1 and 2 using dry (i,ii) and wet (iii,iv) electrodes, respectively.

**Table 1 sensors-24-01810-t001:** Accuracy results from the 5-fold cross-validation during the gesture recognition experiment.

Subject	Condition	Accuracy (%)	Standard Deviation (%)
S1	Dry	97.93	0.75
S2	Dry	95.95	2.97
S1	Wet	98.39	0.96
S2	Wet	98.54	0.66

## Data Availability

All files (including CAD files and PCB schematics) have been uploaded on GitHub (https://github.com/rejinjohnvarghese/Stretchable-HMI-Array accessed on 4 February 2024).

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
