# Peer review of "Design, Fabrication and Evaluation of a Stretchable High-Density Electromyography Array"

_sensors, 2024, doi:10.3390/s24061810_

Round 1
Reviewer 1 Report
Comments and Suggestions for Authors
The manuscript by Varghese et al. present the design, fabrication and evaluation of a stretchable HD-EMG array with potential for human-machine interfaces. Overall, I really liked the manuscript. It provides a simple solution that has multiple end-user applications for many real-world problems. There are just a few comments that the authors may wish to consider.
1. This reviewer feels that the functional implications of this work needs to be highlighted/set-up in the the intro and discussed in the discussion a bit more. As it currently stands (and I appreciate it's a methods approach), it comes across more like a technical manual.
2. I like the approach of using FastICA. However, FastICA can lead to variations between runs which can create problems in terms of consistency. Does this pose a problem with this system?
3. Some additional details on the dynamic wrist gestures are needed. For example, were they cued from a seated position, standing position, or walking? The authors may also want to further comment as these 3 conditions have differing relevance to real-world (naturalistic) conditions, and ultimately, the system's utility. Also, there is evidence that position/postural adjustments can modulate EMG signals (e.g., https://link.springer.com/content/pdf/10.1007/s11062-008-9023-6.pdf), which could impact model accuracy. On the other hand, being able to capture this with this system could be quite beneficial and have clinical implications.
Comments on the Quality of English LanguageThe manuscript is concise, and well-written.
Author Response
Thank you very much for your insightful review. Please find our response to all the concerns raised by the reviewer.
The manuscript by Varghese et al. present the design, fabrication and evaluation of a stretchable HD-EMG array with potential for human-machine interfaces. Overall, I really liked the manuscript. It provides a simple solution that has multiple end-user applications for many real-world problems. There are just a few comments that the authors may wish to consider.
- This reviewer feels that the functional implications of this work needs to be highlighted/set-up in the the intro and discussed in the discussion a bit more. As it currently stands (and I appreciate it's a methods approach), it comes across more like a technical manual.
We have now expanded on the functional implications of the proposed technique both in the introduction (section 1) and discussion (section 6.1) sections, and these expanded sections have been highlighted.
- I like the approach of using FastICA. However, FastICA can lead to variations between runs which can create problems in terms of consistency. Does this pose a problem with this system?
FastICA for EMG decomposition was proposed in (Negro et al. 2016) as an alternative approach to the more known Convolution Kernel Compensation (CKC) method (Holobar & Zazula, 2007). In (Martinez-Valdes, E et al., 2017), the FastICA approach was utilized to show that the same MUs could be tracked in experimental sessions across weeks, proving the consistency of the decomposition approach (and the tracking). Based on the results of these previous studies, we have selected FastICA as a method that should produce consistent and reproducible results across sessions with the same experimental setup.
We have also added the following to the manuscript, in (validation experimental method – 4.1): “FastICA has been previously employed for EMG decomposition, and the decomposition results are consistent in measurements separated by weeks (Martinez-Valdes, E et al., 2017)”.
- Some additional details on the dynamic wrist gestures are needed. For example, were they cued from a seated position, standing position, or walking? The authors may also want to further comment as these 3 conditions have differing relevance to real-world (naturalistic) conditions, and ultimately, the system's utility.
The subjects maintained a seated position while recording the gestures, with the arm parallel to the torso, and the forearm perpendicular to it. The elbow was positioned on a movable support, to prevent excessive fatigue during the experiment sessions. This position was chosen as it could be easily standardised across subjects and would allow for an accurate repetition of the gestures. We have clarified the same in the highlighted part in section 4.2.1
Also, there is evidence that position/postural adjustments can modulate EMG signals (e.g., https://link.springer.com/content/pdf/10.1007/s11062-008-9023-6.pdf), which could impact model accuracy. On the other hand, being able to capture this with this system could be quite beneficial and have clinical implications.
The purpose of this validation experiment was to compare the accuracy of the dry-electrode configuration, against the wet one, and not assess how the recognition accuracy would change according to the subject’s position.
We hope the above responses and changes to the manuscript provide sufficient clarity.
Thank you,
Regards,
Authors
Reviewer 2 Report
Comments and Suggestions for Authors
This study developed a multi-electrode EMG system based on stretchable metal electrodes, which are integrated on a sleeve-like substrate that allows comprehensive recording of EMG signals from the arm or leg through 32 or more electrodes. The authors also proved its effectiveness by using machine learning to categorize the signals with very high accuracy. This study is of great practical value and has prospects for practical application in healthcare and sports monitoring fields, the data collected in this study is complete and the experiment method is illustrated clearly. However, some minor flaws need further clarification.
1. The author claims that the signal-to-noise ratio of wet electrodes is better than that of dry electrodes, here there should be a signal comparison diagram of the corresponding positions of the two specific electrodes, and the specific signal-to-noise ratios should be calculated and the significance level of the hypothesis of this difference value should be stated to remove the effect of random errors.
2, according to the schematic diagram, the stretchable device is assembled by buckled stretchable structures, and the authors should briefly evaluate and characterize the tensile properties of this structure in all directions to illustrate its mechanical properties, such as strain-stress curve.
Author Response
Thank you very much for your insightful comments to improve our manuscript. Please find attached our response and changes to the manuscript.
This study developed a multi-electrode EMG system based on stretchable metal electrodes, which are integrated on a sleeve-like substrate that allows comprehensive recording of EMG signals from the arm or leg through 32 or more electrodes. The authors also proved its effectiveness by using machine learning to categorize the signals with very high accuracy. This study is of great practical value and has prospects for practical application in healthcare and sports monitoring fields, the data collected in this study is complete and the experiment method is illustrated clearly. However, some minor flaws need further clarification.
- The author claims that the signal-to-noise ratio of wet electrodes is better than that of dry electrodes, here there should be a signal comparison diagram of the corresponding positions of the two specific electrodes, and the specific signal-to-noise ratios should be calculated and the significance level of the hypothesis of this difference value should be stated to remove the effect of random errors.
After placing the first grid on the TA muscle, its position was marked using a skin marker. This was done to allow proper placement of the wet-electrode grid afterward, thus providing a fair comparison between the resulting RMS values. Furthermore, to assess the significance of the obtained results, a two-sample t-test was used to compare the RMS values between the two types of electrode grids. For this, the null hypothesis was set to be ’no significant difference between the RMS values of the two groups’, while the alternative hypothesis would be ’significant difference between the RMS values of the two groups’. The p-value to determine the significance was set to be 0.05.
Such result is supported by the two-sample t-test (p = 0.018, <0.05), indicating that there was a significant difference between the RMS values of the two groups, with the RMS of the wet grids being significantly lower than the one of the dry grids. This clarification has also be iterated in the highlighted part in section 5.1.
2, according to the schematic diagram, the stretchable device is assembled by buckled stretchable structures, and the authors should briefly evaluate and characterize the tensile properties of this structure in all directions to illustrate its mechanical properties, such as strain-stress curve.
To clarify the fabrication method, we are using flexible PCBs that can bend considerably, but aren’t stretchable, and therefore, we do not buckle stretchable structures. In the proposed method, we cure the stretchable substrate (such as the silicone rubber material) around the flexible material while the flexible material is in a folded configuration. As detailed in the manuscript, the flexible PCB ranges from 240mm (in its compressed configuration) to 320mm (in its unstretched configuration). This results in equivalent strains in the silicone rubber substrate that is significantly lower than its limits (as documented in commercially available silicone rubber product datasheets).
We have also clarified the same in the highlighted part of the manuscript in section 2.2.3. I hope the above explanation and clarification in the text can be taken into consideration, and not require us to perform tensile tests on the developed sleeve, as this would take considerably longer than the 5 days allocated to us to respond to the reviewers.
We hope these changes and clarifications are sufficient.
Thank you,
Regards,
Authors